# Assessment of the targeted effect of Sijunzi decoction on the colorectal cancer microenvironment via the ESTIMATE algorithm

**Jiaxin Du[1☯], Quyuan Tao[1☯], Ying Liu[1], Zhanming Huang[2], He Jin[1], Wenjia Lin[1], Xinying Huang[1], Jingyan Zeng[3], Yongchang Zhao[2], Lingyu Liu[1], Qian Xu[1], Xue Han[1], Lixia Chen[1], Xin-lin Chen[1‡]*, Yi Wen[2‡]***

**1** School of Basic Medical Science, Guangzhou University of Chinese Medicine, Guangzhou, China, **2** The First Affiliated Hospital, Guangzhou University of Chinese Medicine, Guangzhou, China, **3** Shenzhen Clinical College, Guangzhou University of Chinese Medicine, Guangzhou, China

☯ These authors contributed equally to this work.
‡ XC and YW also contributed equally to this work.
* chenxlsums@126.com (XC); 20172101031@stu.gzucm.edu.cn (YW)

**Data Availability Statement:** All relevant data are within the manuscript and its Supporting Information files.

## Abstract

### Objective

Sijunzi decoction (SJZD) was used to treat patients with colorectal cancer (CRC) as an adjuvant method. The aim of the study was to investigate the therapeutic targets and pathways of SJZD towards the tumor microenvironment of CRC via network pharmacology and the ESTIMATE algorithm.

### Methods

The ESTIMATE algorithm was used to calculate immune and stromal scores to predict the level of infiltrating immune and stromal cells. The active targets of SJZD were searched in the Traditional Chinese Medicine Systems Pharmacology Database and Analysis Platform (TCMSP) and UniProt database. The core targets were obtained by matching the differentially expressed genes in CRC tissues and the targets of SJZD. Then, GO, KEGG and validation in TCGA were carried out.

### Results

According to the ESTIMATE algorithm and survival analysis, the median survival time of the low stromal score group was significantly higher than that of the high stromal score group ($P$ = 0.018), while the patients showed no significant difference of OS between different immune groups ($P$ = 0.19). A total of 929 genes were upregulated and 115 genes were downregulated between the stromal score groups (|logFC| > 2, adjusted $P$ < 0.05); 357 genes were upregulated and 472 genes were downregulated between the immune score groups. The component-target network included 139 active components and 52 related targets. The core targets were *HSPB1*, *SPP1*, *IGFBP3*, and *TGFB1*, which were significantly

**Funding:** This study was funded by the National Natural Science Foundation of China (81774451), the Science Program for Overseas Scholars (Xinhuo plan) of Guangzhou University of Chinese Medicine (XH20190102), 2020 National Innovation and Entrepreneurship Program for Chinese College Students (202010572017), Guangzhou University of Chinese Medicine College Student Innovation and Entrepreneurship Project (202010572201), and Guangzhou Science and technology project (201707010319). The funders had no role in study design, data collection and analysis, decision to publish, or preparation of the manuscript.

**Competing interests:** The authors have declared that no competing interests exist.

associated with poor prognosis in TCGA validation. GO terms included the response to hypoxia, the extracellular space, protein binding and the TNF signaling pathway. Immunoreaction was the main enriched pathway identified by KEGG analysis.

## Conclusion

The core genes (*HSPB1*, *SPP1*, *IGFBP3* and *TGFB1*) affected CRC development and prognosis by regulating hypoxia, protein binding and epithelial-mesenchymal transition in the extracellular matrix.

## Introduction

Cancer development and progression are complex processes caused by the accumulation of genetic modifications in cancer cells and influenced by the surrounding microenvironment. Cancer cells recruit vascular and stromal components (including immune cells, fibroblasts, cytokines, and the extracellular matrix that surrounds them) to the tumor microenvironment (TME), and the activated TME in turn modifies the malignant behaviors of cancer cells [1]. Numerous studies have indicated that infiltrating immune cells in the TME fail to execute anti-tumor functions but interact intimately with tumor cells to promote oncogenesis and progression. Both innate and adaptive immune cells are involved in this cross-talk [2].

Colorectal cancer (CRC) is a kind of digestive tract tumor, mostly affecting the colon, that is influenced by multiple factors, such as genetics, environment, and diet. With changes in dietary structure and the deterioration of the natural environment, the incidence and mortality of CRC continue to rise [3–5]. In 2018, there were 18.1 million new cancer cases and 9.6 million cancer-related deaths worldwide. China has the highest cancer incidence and mortality rates in the world, and CRC ranks fourth in the incidence and second in the mortality rate among malignant tumors in China [6]. At present, surgical resection is still the only treatment that can completely cure CRC. However, since the initial symptoms of patients with CRC are not obvious, most patients are diagnosed at the late stage when surgery is not an option [7]. Moreover, up to 15% of patients develop major complications, and postoperative recurrence and metastasis are common [8]. Although the chemotherapy combination of 5-fluorouracil with oxaliplatin has been shown to reduce the rates of metastasis and recurrence in advanced CRC [9, 10], long-term chemoradiotherapy, like many chemotherapy-related strategies, has adverse effects, such as organ damage, myelosuppression, cumulative neuropathy, severe reduction in the quality of life of patients and drug resistance [11, 12].

Traditional Chinese medicine (TCM) plays an indispensable role in the integrative treatment of advanced CRC and has been widely used in China and other parts of Asia. It has been widely reported that TCM could prolong the survival time and improve the quality of life of patients with CRC by preventing tumorigenesis, suppressing tumor growth, and reducing metastasis and recurrence [13–16]. Moreover, as an adjunct therapy, TCM could increase the sensitivity and alleviate the side effects of chemotherapy or radiotherapy, improve immunity, lessen the damage induced by conventional treatments, and ameliorate bone marrow suppression [11, 17–20]. Sijunzi decoction (SJZD), which consists of Panax Ginseng C. A. Mey, Atractylodes Macrocephala Koidz, Poria Cocos Wolf and licorice, is a classical prescription for curing temper deficiency syndrome in TCM. It has long been widely used in the clinic to treat CRC since it combats vomiting, diarrhea and nausea to restore homeostasis in the digestive tract [21–25]. Emerging evidence has shown that SJZD is beneficial for advanced CRC

treatment, as it improves the quality of life of patients with Qi deficiency syndrome [26]. However, previous studies mainly focused on the differentially expressed genes (DEGs) between tumor tissues and nontumor tissues, but these studies were often insufficient and limited because they ignored the influence of TCM on the TME. Here, we predicted the levels of infiltrating stromal and immune cells by algorithmically calculating the stromal and immune score. The therapeutic targets of SJZD and the DEGs between different infiltration levels were intersected, and the core genes affected the TME of CRC were found. The result will provide the TME of CRC with a novel auxiliary therapeutic perspective.

## Materials and methods

### Data mining and processing of the CRC expression profile chip

The Gene Expression Omnibus (GEO, https://www.ncbi.nlm.nih.gov/geo/) database was generated and maintained through the National Center for Biotechnology Information (NCBI) Gene Expression database. The GEO includes high-throughput gene expression data submitted by research institutions around the world [27]. The datasets were considered to be appropriate for further analysis according to two criteria: (1) studies with CRC tissue samples and (2) studies with clinical prognostic data available for the samples. Based on these criteria, GSE31595 [28] and GSE17536 [29] were evaluated.

Potential variables and heterogeneity are commonly considered to be major sources of variability and bias in high-throughput experiments [30]. The obtained sample data were prone to error due to the small sample size and differences in sample handling and study design. Therefore, all samples from both datasets were integrated to significantly increase the sample size. Then, the scale function was used in R 4.0.2 software to normalize and standardize the samples.

### Evaluation of the infiltration degree of stromal cells and immune cells based on the Estimation of STromal and Immune cells in MAlignant Tumor tissues using Expression data (ESTIMATE) algorithm

The ESTIMATE algorithm [31], which utilizes the unique properties of the transcriptional profiles of cancer samples to infer tumor cellularity and different degrees of infiltration of tumor and normal cells, was applied. To identify specific signatures related to the infiltration of stromal and immune cells, immune and stromal scores were calculated with the ESTIMATE algorithm, and these scores reflected the predicted level of infiltrating stromal and immune cells. In this study, the ESTIMATE package (https://CRAN.R-project.org/package=EstimateGroupNetwork) in R 4.0.2 was used to perform ESTIMATE scoring of the CRC samples. Besides, in order to compare the variation trends of immune and stromal score at different pathological stages, box plot was used to visualized the distribution characteristics of immune and stromal score at different American Joint Committee on Cancer (AJCC) stages.

### Prognostic analysis of the ESTIMATE score

Based on the results of the ESTIMATE score (stromal score or immune score), the 214 samples were divided into two groups (high: ESTIMATE score $\geq$ median, and low: ESTIMATE score $<$ median). To determine the validity of the ESTIMATE score, a prognosis analysis was utilized to analyze the relationship between the ESTIMATE score and the survival time of the patients with CRC, and survival curves were evaluated by Kaplan–Meier method. In the survival analysis, the clinical data of all patients were obtained from the previously downloaded data sets. Overall survival (OS) was defined as the time from the beginning of random

assignment to death (for patients still alive at the end of the study, the time of death was the end of follow-up).

## Gene set enrichment analysis

To explore the significant biological pathway of different infiltration levels in the TME of CRC, gene set enrichment analysis (GSEA, http://software.broadinstitute.org/gsea/index.jsp) was performed in the immune and stromal group respectively. The expression matrix and grouping information (stromal high vs stromal low, immune high vs immune low) of 214 samples were used as input data. GSEA v4.1.0 software was used to evaluate the enrichment of biological pathway with "weighted" enrichment statistics. Enriched gene sets with false discovery rate (FDR) < 0.25, |normalized enrichment score (NES)| > 1, and nominal $P < 0.05$ were regarded as being statistically significant [32].

## Collection of active ingredients

The Traditional Chinese Medicine Systems Pharmacology Database and Analysis Platform (TCMSP, http://tcmspw.com/tcmsp.php/) [33] was used to retrieve active compounds of SJZD based on absorption, distribution, metabolism, and excretion (ADME) [34]. This is a unique pharmacology platform devised for TCM to capture the disease-herb-target relationship.

As oral administration was the chief route of drug delivery of TCM, oral bioavailability (OB) was set as the major parameter in active constituent retrieval [35]. In this study, OB≥30% and drug likeness (DL)≥0.18 were used as the standards to screen the effective active compound of SJZD [36].

## Drug target screening and core target gene acquisition of SJZD

The effective active ingredients acquired from TCMSP in the previous step were further searched to obtain the corresponding target proteins. The standardized genes were retrieved through the UniProt database (https://www.Uniprot.org/) [37]. The active genes of Panax Ginseng C. A. Mey, Atractylodes Macrocephala Koidz, Poria Cocos Wolf and licorice were sorted out and introduced into Cytoscape V3.7.2 to establish the "compound-target" network diagram to visualize the overall situation of drug compounds and targets. The network topology analysis plug-in CytoNCA [38] in Cytoscape V3.7.2 was used to obtain the core compound and target gene of SJZD with a degree of freedom > 34 (twice the median). "Degree" reflects the number of links with each node in the network and the interaction relationship between nodes.

## Screening of core genes related to SJZD in CRC

The stromal scores and immune scores of the samples were used as the grouping parameters for differential expression analysis. Stromal scores were classified into two groups: high stromal score group (HSS) and low stromal score group (LSS). Similarly, high immune score group (HIS) and low immune score group (LIS) were obtained. The DEGs of stromal groups or immune groups based on the TME were obtained using differential expression analysis (|Log2FC| > 2, adjusted $P \leq 0.05$) according to the score groups (HSS vs LSS, HIS vs LIS) in the R package limma [39]. The therapeutic targets of SJZD for CRC obtained previously were intersected with these DEGs to obtain the core target genes of SJZD in CRC.

## Analysis of core target genes related to SJZD in CRC

To explore the specific pathways and biological processes of the core target genes involved, gene ontology (GO) enrichment analysis and Kyoto Encyclopedia of Genes and Genomes

(KEGG) pathway analysis were performed based on the clusterProfiler package [40]. Protein-protein interaction (PPI) were analyzed and constructed using the Search Tool for the Retrieval of Interacting Genes (STRING) database (https://string-db.org/cgi/input.pl) [41], which could retrieve further protein interactions of the core targets. The PPI network was visualized in Cytoscape V3.7.2. Topological analysis was carried out, and the degree value was used as a reference to determine the important core targets. To investigate the effect of these core target genes on the prognosis of CRC patients, a survival analysis was performed using the clinical data from GEO database (GSE31595 and GSE17536). In addition, considering the limited number of patient samples, we performed further survival analysis validation by using 530 colon adenocarcinoma (COAD) samples from the TCGA database.

Survival curves were drawn to screen core target genes with prognostic value ($P \leq 0.05$). The core target genes with prognostic value were called microenvironment-related prognostic genes.

## Results

### Immune scores and stromal scores were associated with CRC stage

First, the gene expression profiles and clinical characteristics of all 214 patients pathologically diagnosed with CRC were obtained from the GEO database. Among them, 103 patients were female (48.1%), and 111 patients were male (51.9%). The pathological classification of AJCC stage revealed 24 (11.2%) cases of stage I disease, 77 (36.0%) cases of stage II disease, 74 (34.6%) cases of stage III disease and 39 (18.2%) cases of stage IV disease.

According to the ESTIMATE algorithm, the ranges of stromal scores and immune scores were −323.92~5403.97 and -859.48~6701.10, respectively. According to the overall trend of the box plot, the stromal and immune score generally increased with the progression of AJCC stage (Fig 1A and 1B). Just in terms of the median value, the median of stage IV is lower than that of stage III, but still higher than that of stage II and stage I (Fig 1A). Similarly, the rank order of immune score for CRC stages was stage III > stage IV > stage II > stage I (Fig 1B), indicating that immune scores and stromal scores based on the ESTIMATE algorithm were positively correlated with the pathological diagnosis and clinical stage to a certain extent.

To assess the potential relationships between overall survival (OS) and immune scores and stromal scores, all the patients were divided into high score groups ($\geq$ median ESTIMATE score) and low score groups ($<$ median ESTIMATE score). Kaplan-Meier survival curves indicated that the median survival time of the low stromal score group was significantly higher than that of the high stromal score group (Fig 1D, $P = 0.018$), while the patients showed no significant difference of OS between different immune groups (Fig 1C, $P = 0.19$)

### Results of gene set enrichment analysis (GSEA)

In the C5 enrichment analysis results, 658 gene sets were significantly enriched in the high score group; there were 10 enriched gene sets in the low score group. According to the immunization score-based grouping, 678 and 1 gene sets were significantly enriched for the high group and low group, respectively (S1 Appendix).

The results of C5 enrichment analysis in the high stromal score group showed that the infiltration of stromal cells in the TME of CRC might be related to the following biological functions: cytokine mediated signaling pathway, G protein coupled receptor signaling pathway coupled to cyclic nucleotide second messenger, B cell activation, side of membrane, and adaptive immune response (Fig 2B). According to the GSEA results, cytokine mediated signaling pathway was the most enriched biological function in high stromal score group (Fig 2A,

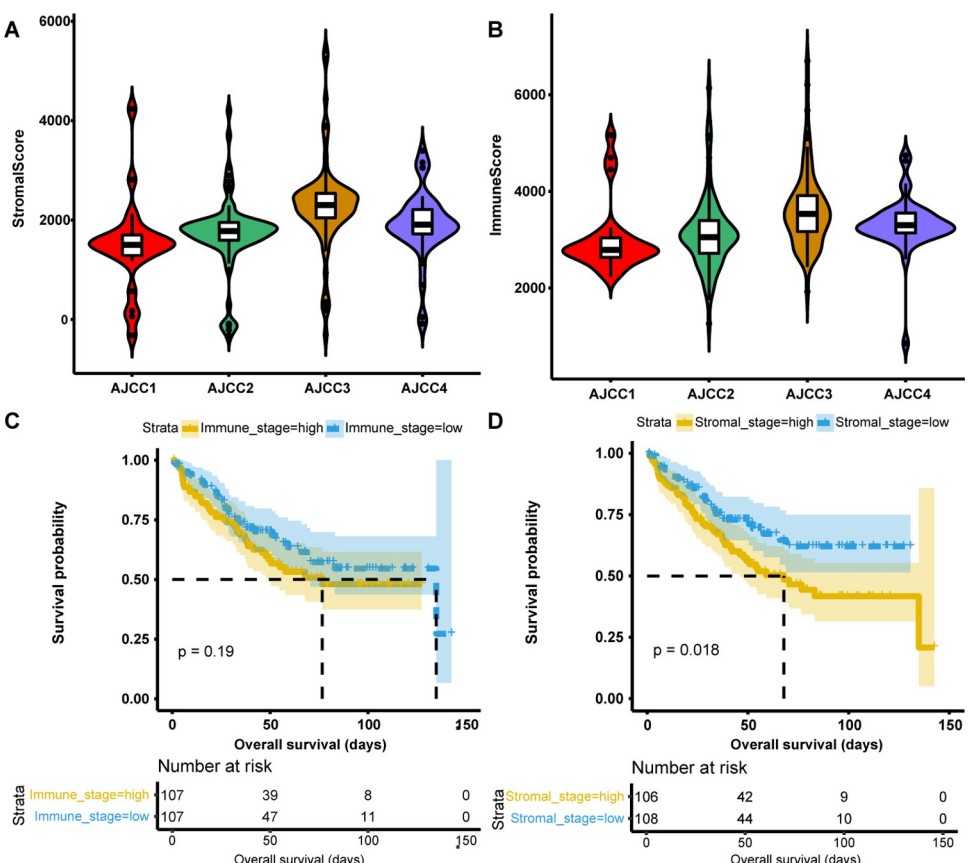

**Fig 1. Results of correlation analysis.** (A) Association between stromal score and CRC AJCC stage. (B) Association between the immune score and CRC AJCC stage. (C) Correlation between the stromal score group and the survival time of patients with CRC. (D) Correlation between the immune score group and the survival time of patients with CRC.

S1 Appendix), which could modulate the ability of cancer cell to grow, proliferate, invade, and metastasize [42].

The results of C5 enrichment analysis in the high immune score group suggested that the infiltration of immune cells in the TME of CRC might be related to the following biological functions: amide binding, response to wounding, endosomal part, calcium ion transmembrane transport, and endocytic vesicle (Fig 2D). The biological function of amide binding was of the highest NES in the high immune score (Fig 2C, S1 Appendix), which indicated that this gene set might regulate the immune response in patients with CRC by inducing related amide binding, that was consistent with the previous research [43].

## DEGs

The DEGs of HSS vs LSS and HIS vs LIS were integrated to obtain the union of 1557 DEGs (S2 Appendix). Heat map showed the different expression levels of the top 50 genes between the stromal score groups and immune score groups (Fig 3A and 3B). The results showed that 357 genes were upregulated and 472 genes were downregulated between the immune score groups ($|logFC| > 2$, adjusted $P < 0.05$, Fig 3C); 929 genes were upregulated and 115 genes were downregulated between the stromal score groups ($|logFC| > 2$, adjusted $P < 0.05$, Fig 3D).

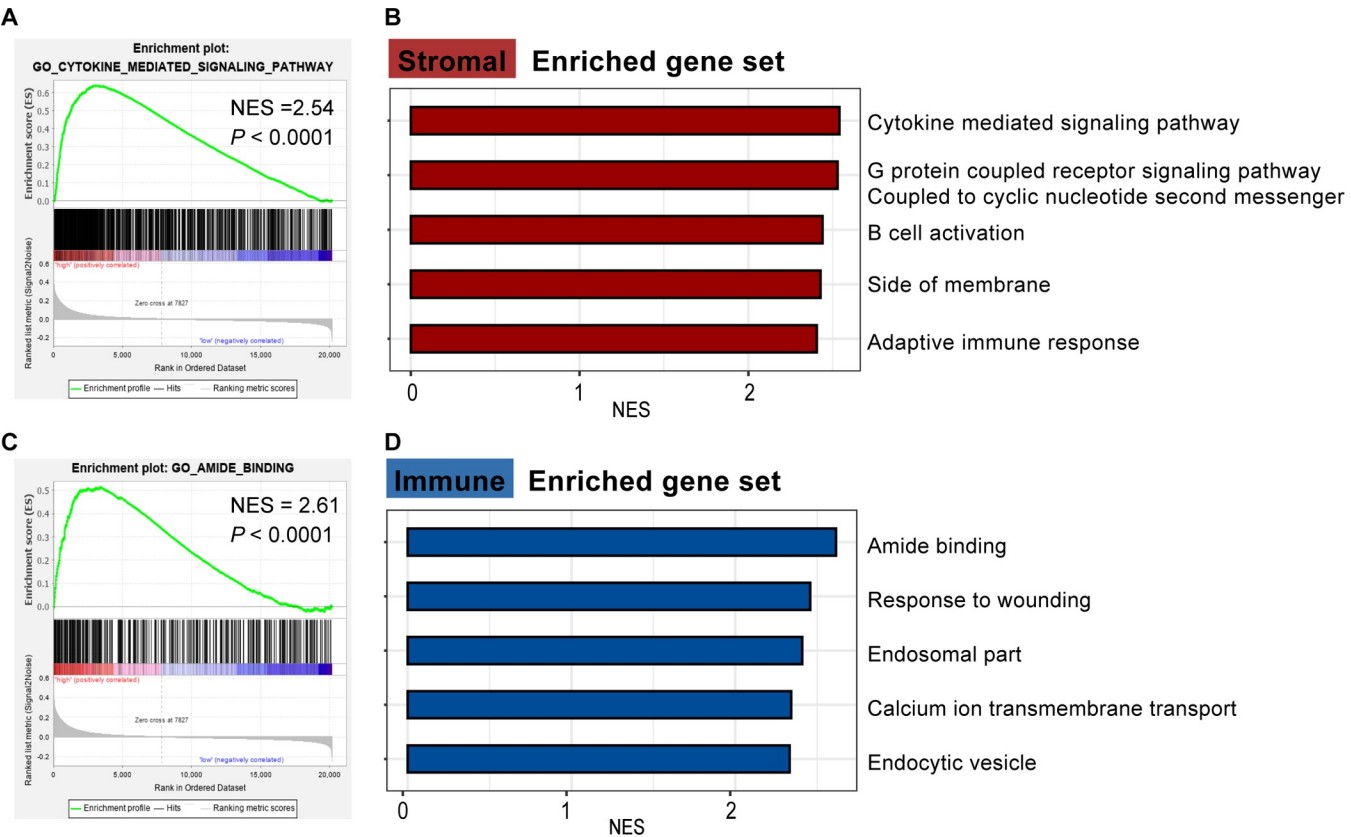

**Fig 2. Estimate score C5 enrichment analysis results based on the ESTIMATE algorithm.** (A) GSEA of top one pathway enriched for high stromal group (NES = 2.54, nominal *P* < 0.0001). (B) The top5 significant biological pathways in high stromal score group (sort by NES). (C) The enrichment plot of amide binding (NES = 2.61, nominal *P* < 0.0001). (D)The top5 significant biological pathways in high immune score group (sort by NES).

### Network pharmacological analysis indicated the importance of each gene in SJZD treatment

Through the TCMSP database, a total of 139 active compounds that met the screening conditions were obtained. Some active ingredients of the four herbs of SJZD were shown in Table 1. Complete information on the active ingredients of SJZD can be found in S1 Table. The official human genes corresponding to the 230 target proteins were retrieved by using UniProtKB. Compound-target networks were constructed by using Cytoscape (Fig 4A). The degree value represents the degree of connection between the targets, and a high degree of connection indicates great importance. The top 12 targets are shown in detail in Table 2. In the network diagram, the square nodes are drug targets, and the red nodes are compounds of SJZD. The larger the node is, the greater the importance of the drug component or target is.

52 core target genes were obtained after the intersection of 1557 DEGs and 230 protein-coding genes (Fig 4B). These 52 genes were utilized to obtain a PPI network diagram (Fig 4C). Genes with larger nodes and deeper colors have higher degree values.

### Biological pathways related to the target DEGs

The functional enrichment analysis of the 52 core target genes showed that they were mainly enriched in the signal transduction and expression regulation in TME of CRC. Among the enriched GO terms, there were 153 significantly enriched biological processes (BP), 17

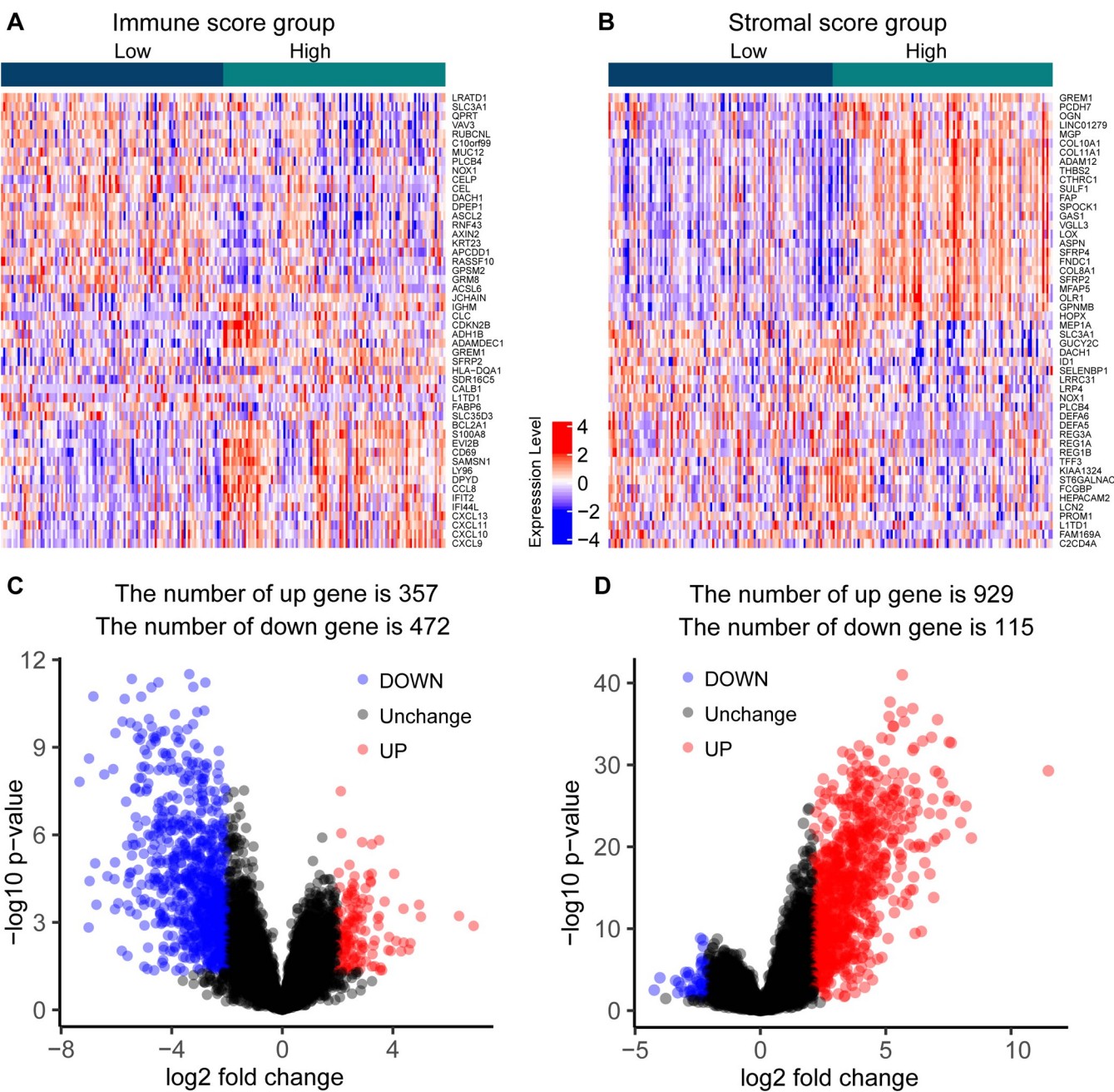

**Fig 3. The expression profiles and differential expression analysis results of the top 50 characteristic genes between the high and low stromal and immune score groups.** (A) The expression heat map for the top 50 DEGs based on immune score grouping. (B) The expression heatmap for the top 50 DEGs based on stromal score grouping. (C) Volcano maps based on differential expression analysis of immune score groups. (D) Volcano maps based on differential expression analysis of stromal score groups.

significantly enriched cellular components (CC) and 25 significantly enriched molecular functions (MF) (*P* < 0.05). The main enriched GO terms included response to hypoxia and inflammatory response (Fig 5A), extracellular space, extracellular region and extracellular exosome (Fig 5B), and protein binding, receptor binding, enzyme binding and serine−type endopeptidase activity (Fig 5C). In addition, all pathways identified as enriched by the KEGG analysis

**Table 1. Basic information for some active components of SJZD.**

| Mol ID | Chemical component | OB (%) | DL | Herb |
|---|---|---|---|---|
| MOL000273 | (2R)-2-[(3S,5R,10S,13R,14R,16R,17R)-3,16-dihydroxy-4,4,10,13,14-pentamethyl-2,3,5,6,12,15,16,17-Octahydro-1H-cyclopenta[a]phenanthren-17-yl]-6-methylhept-5-enoic acid | 30.93 | 0.81 | Poria Cocos Wolf. |
| MOL000275 | Trametenolic acid | 38.71 | 0.80 | |
| MOL000276 | 7,9(11)-Dehydropachymic acid | 35.11 | 0.81 | |
| MOL000279 | Cerevisterol | 37.96 | 0.77 | |
| MOL001484 | Inermine | 75.18 | 0.54 | licorice |
| MOL001792 | DFV | 32.76 | 0.18 | |
| MOL000211 | Mairin | 55.38 | 0.78 | |
| MOL002879 | Diop | 43.59 | 0.39 | Panax Ginseng C. A. Mey. |
| MOL000449 | Stigmasterol | 43.83 | 0.76 | |
| MOL000358 | β-Sitosterol | 36.91 | 0.75 | |
| MOL003648 | Inermin | 65.83 | 0.54 | |
| MOL000022 | 14-Acetyl-12-senecioyl-2E,8Z,10E-atractylentriol | 63.37 | 0.3 | Atractylodes Macrocephala Koidz. |
| MOL000033 | (3S,8S,9S,10R,13R,14S,17R)-10,13-dimethyl-17-[(2R,5S)-5-propan-2-yloctan-2-yl]-2,3,4,7,8,9,11,12,14,15,16,17-dodecahydro-1H-cyclopenta[a]phenanthren-3-ol | 36.23 | 0.78 | |
| MOL000049 | 3β-Acetoxyatractylone | 54.07 | 0.22 | |
| MOL000072 | 8β-Ethoxy atractylenolide III | 35.95 | 0.21 | |

were associated with TNF signaling pathway, transcriptional misregulation in cancer and pathways in cancer (Fig 5D).

## Survival analysis and validation

Survival analysis of the 52 core target genes showed that *HSPB1*, *COL3A1*, *SPP1*, *OLR1*, *IGFBP3* and *TGFB1* were associated with adverse prognosis of the 214 GEO CRC patients. Validation of 530 TCGA samples further confirmed that *HSPB1*, *SPP1*, *IGFBP3* and *TGFB1* were indeed associated with poor prognosis. These 4 genes were identified as the main targets of SJZD in the CRC TME, called the core prognostic targets of SJZD for patients with CRC. The survival curves were shown in Figs 6 and 7.

## Discussion

Network pharmacology analysis clarified a potential mechanism underlying the role of SJZD in the CRC TME and indicated that *HSPB1*, *SPP1*, *IGFBP2* and *TGFB1* could be potential therapeutic targets in the CRC TME. GO and KEGG analysis of 52 core genes showed that they were mainly enriched in the response to hypoxia, the extracellular space, protein binding and the TNF signaling pathway. After a comprehensive analysis of these results, we supposed that the effect of SJZD on the microenvironment of CRC might be realized via the following three mechanisms: SJZD remodels the TME and prolongs the survival of patients with CRC by altering the expression of *SPP1*, *TGFB1A* and *IGFBP3*, attenuating tumor hypoxia, decreasing the reactive oxygen species (ROS) level and enhancing the sensitivity to drugs. Hypoxia is a hallmark of the TME and plays roles in metastasis, therapy resistance and adverse clinical outcomes [44]. The prognosis of CRC is significantly related to the hypoxia response, immunity, and combined treatment response, and CRC commonly exhibits an immunosuppressive phenotype [45]. Epithelial-mesenchymal transition (EMT) is promoted by hypoxia to facilitate invasion, migration and extravasation of CRC cells [44]. ① *SPP1* is considered to be related to

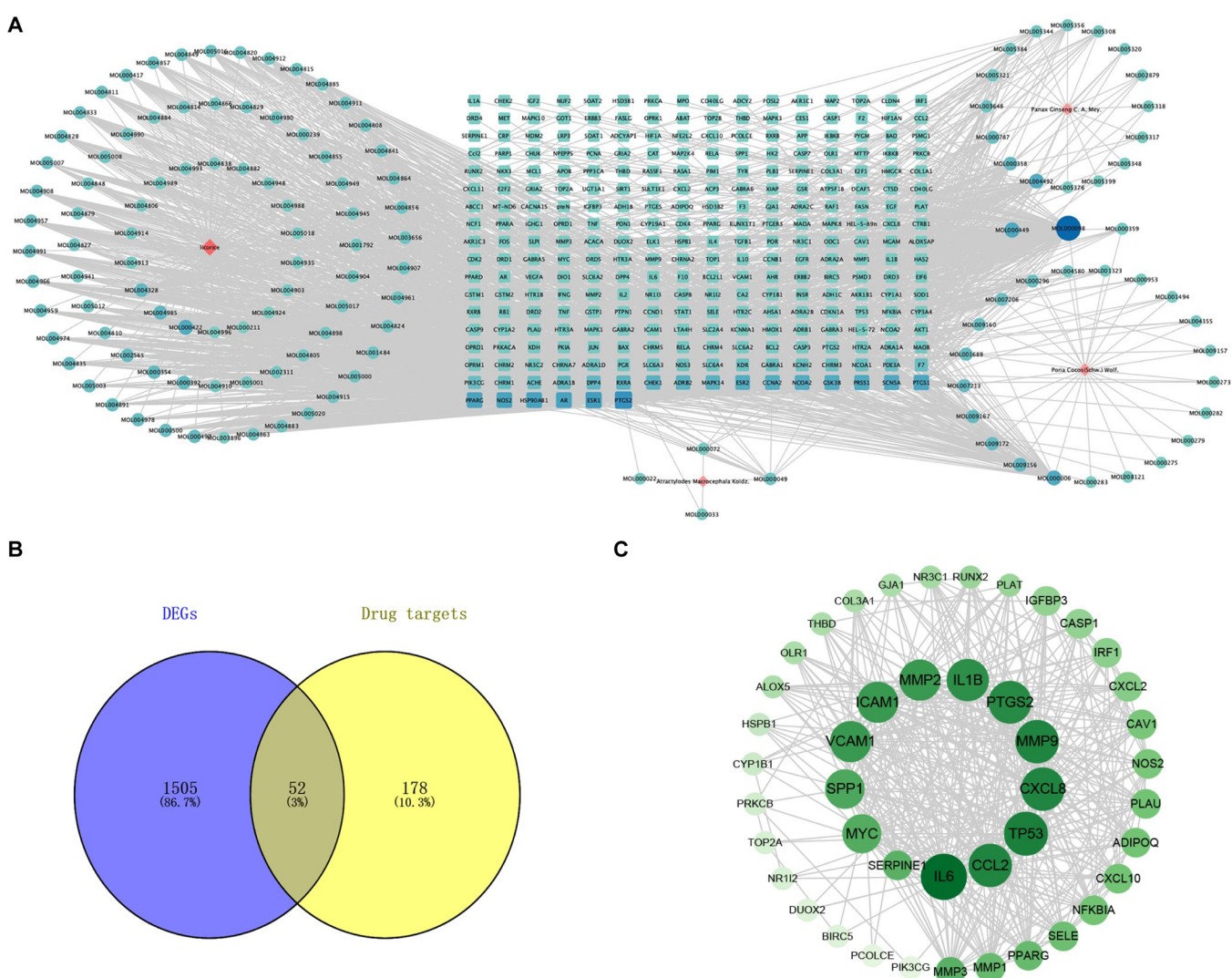

**Fig 4. Network pharmacology analysis of SJZD.** (A) Drug-active ingredient-target network diagram. The blue nodes are drug targets, and the red nodes are compounds of SJZD. (B) Intersection of 230 therapeutic targets and 1557 DEGs related to SJZD. (C) PPI network diagram of the 52 core genes. The larger nodes in the inner ring represented the most important hub nodes. The smaller nodes in the outer ring represent the other nodes (several nodes with degrees of 1 were deleted).

**Table 2. A partial information table for the core targets.**

| Targets | Degree | Targets | Degree |
|---|---|---|---|
| PTGS2 | 86.0 | PRSS1 | 57.0 |
| ESR1 | 77.0 | GSK3B | 56.0 |
| NOS2 | 67.0 | SCN5A | 53.0 |
| PPARG | 67.0 | ESR2 | 53.0 |
| AR | 67.0 | CCNA2 | 53.0 |
| HSP90AB1 | 64.0 | PTGS1 | 52.0 |

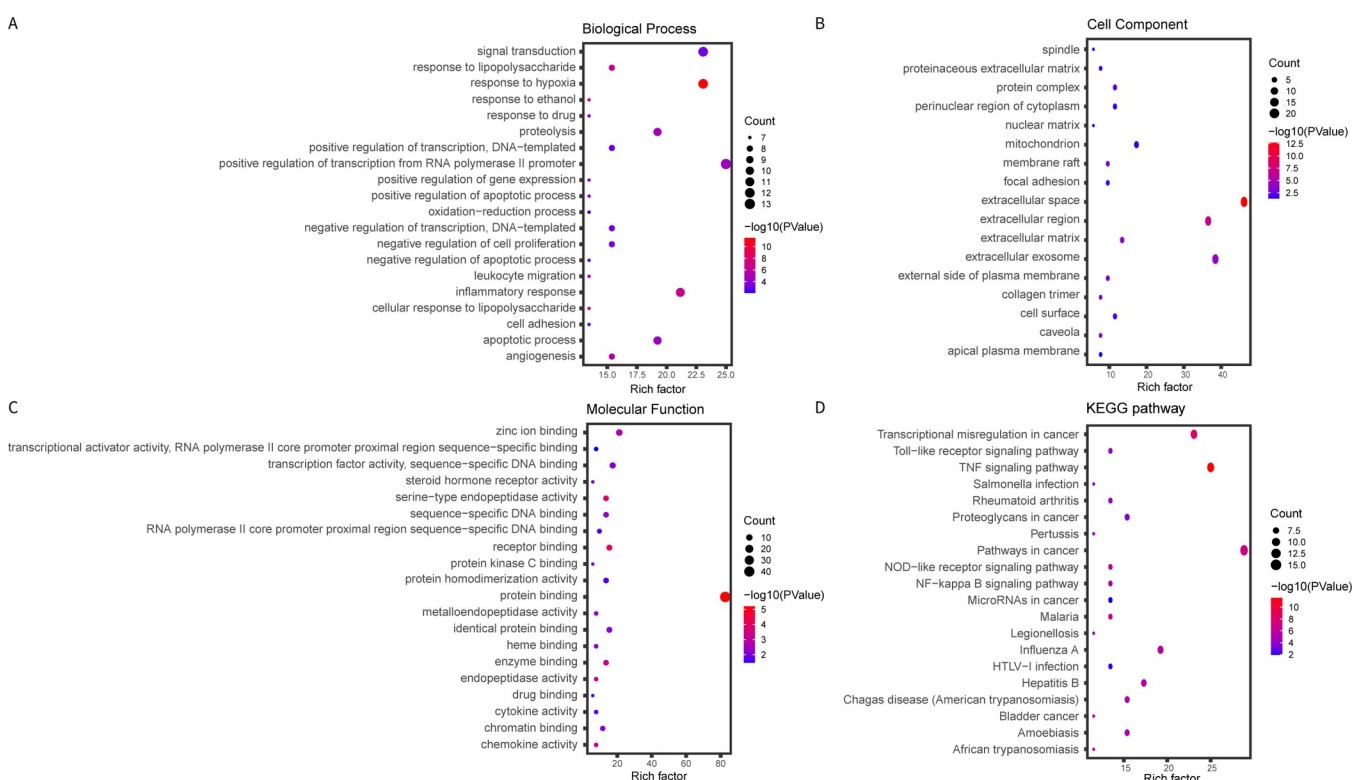

**Fig 5. GO (BP, MF, CC) enrichment analysis and KEGG pathway enrichment analysis of 52 nodes.**

hypoxia, whereas the differential expression of *SPP1* might indicate a hypoxic response. Osteopontin, encoded by *SPP1*, promotes the carcinogenesis, disease progression and recurrence of CRC [46, 47]. ② The hypoxia-associated gene *TGFB1A* is upregulation in the extracellular space, and this upregulation is related to immune evasion and immunological rejection [48, 49]. ③ The *IGFBP3* gene is induced under hypoxic conditions; it regulates cell proliferation, senescence, apoptosis, and EMT and promotes the upregulation of a major cell surface receptor of hyaluronic acid, CD44 (CD44H). *IGFBP3* inhibits ROS cytotoxicity to promote CD44 expression in the hypoxic microenvironment [49–52].

SJZD might inhibit EMT by altering the expression of *TGFB1* to suppress the progression and influence the prognosis of CRC. The cellular EMT program is crucial for malignant progression; EMT is a coordinated process involving recombination of cell-extracellular matrix, protein binding, and regulation of drug resistance and tumor cell metastasis [53–55]. The hub gene *TGFB1* has been shown to efficiently activate EMT. When EMT is activated, the intracellular and extracellular structures and interactions are altered; these changes include the disruption of cell-cell junctions and reorganization of the extracellular matrix [56]. EMT endows cancer cells with tumor invasion and metastasis capabilities [57]. Research has supported that the EMT program, which is mediated by EMT-related transcription factors, could transform nonmetastatic primary carcinoma cells into invasive and metastatic cells [58].

SJZD affects protein binding in the CRC TME by altering the expression of *HSPB1*, *IGFBP-3* and *SPP1*. CRC tumor cells have high metastatic ability due to their low adhesion properties [59]. Protein binding mediates the response to intestinal injury to fuel regeneration [60]. ①

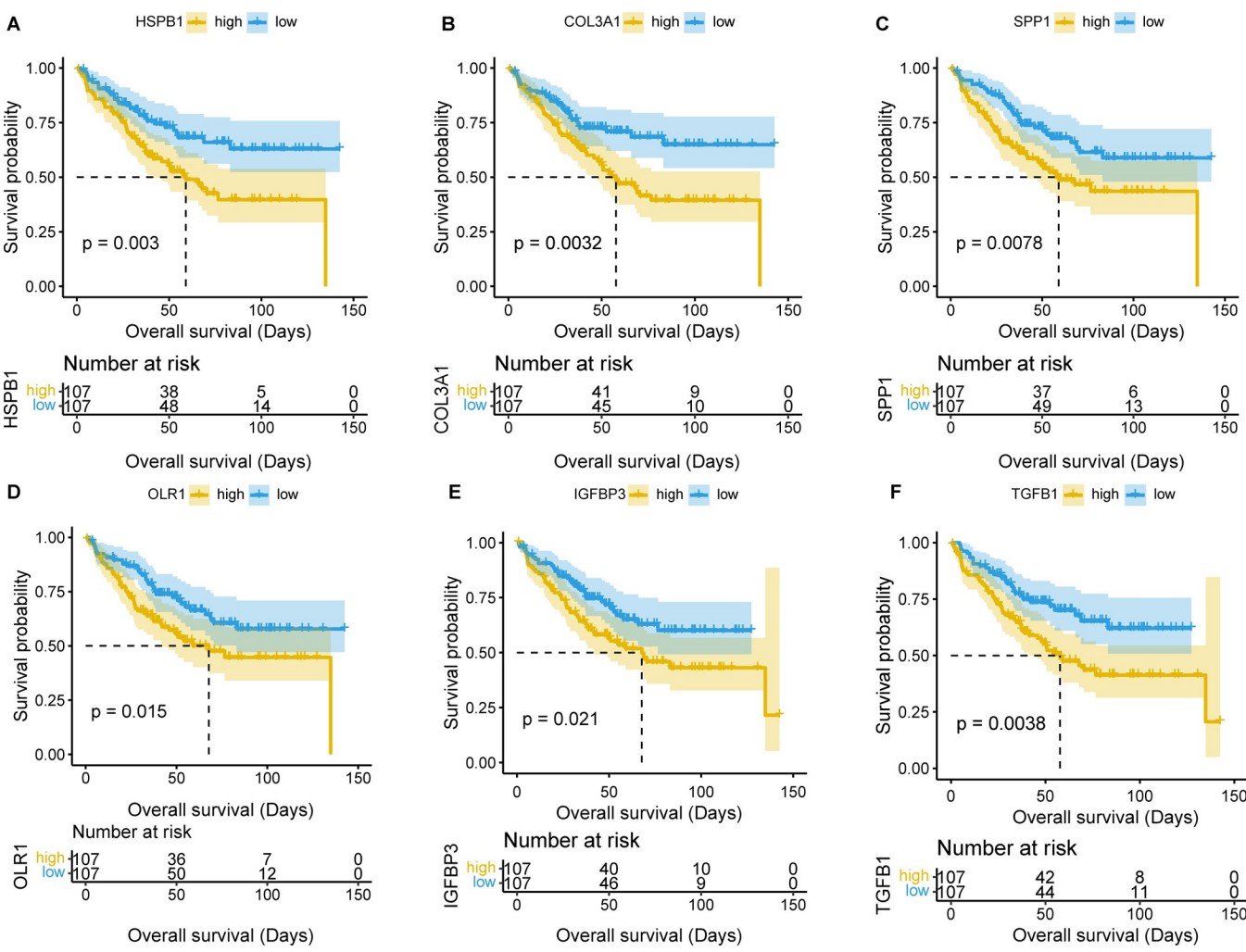

**Fig 6. Survival analysis results for the 52 core target genes (performed in 214 CRC samples from GEO).**

*HSPB1*, also called *HSP27*, is a marker of metastatic tumors that partitions damaged proteins and prevents protein aggregation [61]. Overexpression of *HSPB1* is associated with a poor prognosis in CRC [62]. ② Research has shown that *IGFBP-3* cannot enter cells; thus, it interacts with surface proteins to accelerate tumor survival and invasion [63]. ③ *SPP1* alters the normal expression pattern of intestinal epithelial cells by binding the IRF8 protein, and the expression of *SPP1* is related to intestinal diseases such as ulcerative colitis [47, 64].

Nevertheless, there are still limitations and deficiencies in this study, such as the small sample size. We will explore the specific mechanism of these core genes in further research to provide a more accurate analysis of how the CRC TME affects prognosis.

## Conclusions

In conclusion, our study explored core genes involved in the therapeutic mechanism of SJZD in the CRC TME. The results confirmed that the core genes *HSPB1*, *SPP1*, *IGFBP3* and *TGFB1* affected CRC development and prognosis by regulating hypoxia, protein binding and EMT in the extracellular matrix, which presents a new idea for the treatment of CRC.

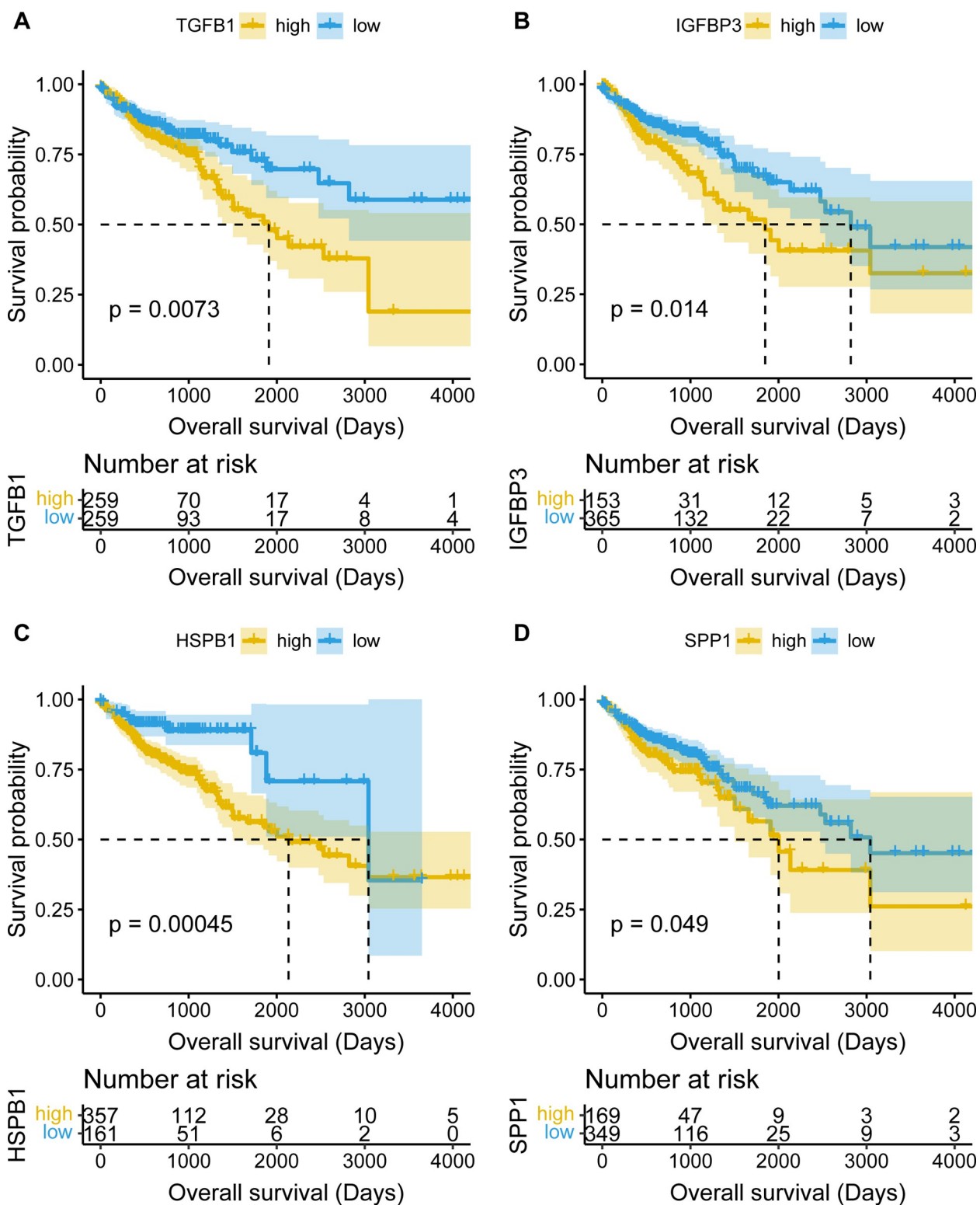

**Fig 7. Verification results of survival analysis (performed in 530 COAD samples from TCGA).**

## Supporting information

**S1 Table. Basic information of some active components of SJZD.**
(DOCX)

**S1 Appendix. The details of the GSEA results.**
(XLSX)

**S2 Appendix. The list of the DEGs.**
(XLSX)

## Author Contributions

**Conceptualization:** Jiaxin Du, Jingyan Zeng.

**Data curation:** Jiaxin Du, Quyuan Tao, Xinying Huang.

**Formal analysis:** Jiaxin Du, Ying Liu, Wenjia Lin.

**Methodology:** Quyuan Tao, Jingyan Zeng, Xin-lin Chen, Yi Wen.

**Project administration:** Quyuan Tao, Xin-lin Chen, Yi Wen.

**Supervision:** Zhanming Huang, He Jin, Yongchang Zhao, Lingyu Liu, Qian Xu, Xue Han, Lixia Chen, Xin-lin Chen, Yi Wen.

**Validation:** Quyuan Tao, Zhanming Huang, He Jin, Jingyan Zeng, Yongchang Zhao, Lingyu Liu, Qian Xu, Xue Han, Lixia Chen.

**Visualization:** Quyuan Tao, Ying Liu, Wenjia Lin, Xinying Huang, Jingyan Zeng.

**Writing – original draft:** Jiaxin Du, Quyuan Tao, Ying Liu, Wenjia Lin, Xinying Huang.

**Writing – review & editing:** Jiaxin Du, Quyuan Tao, Ying Liu, Zhanming Huang, He Jin, Jingyan Zeng, Yongchang Zhao, Lingyu Liu, Qian Xu, Xue Han, Lixia Chen, Xin-lin Chen, Yi Wen.

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
