## [Decision Letter · Decision Letter 0]

6 Dec 2021

PONE-D-21-21615

Assessment of the targeted effect of Sijunzi decoction on the colorectal cancer microenvironment via the ESTIMATE algorithm

PLOS ONE

Dear Dr. Chen,

Thank you for submitting your manuscript to PLOS ONE. After careful consideration, we feel that it has merit but does not fully meet PLOS ONE’s publication criteria as it currently stands. Therefore, we invite you to submit a revised version of the manuscript that addresses the points raised during the review process.

We look forward to receiving your revised manuscript.

Kind regards,

Ilya Ulasov, Ph.D

Academic Editor

PLOS ONE

Journal Requirements:

Additional Editor Comments (if provided):

Reviewers' comments:

Reviewer's Responses to Questions

**Comments to the Author**

1. Is the manuscript technically sound, and do the data support the conclusions?

Reviewer #1: Yes

Reviewer #2: Partly

2. Has the statistical analysis been performed appropriately and rigorously? 

Reviewer #1: Yes

Reviewer #2: No

3. Have the authors made all data underlying the findings in their manuscript fully available?

Reviewer #1: Yes

Reviewer #2: No

4. Is the manuscript presented in an intelligible fashion and written in standard English?

Reviewer #1: Yes

Reviewer #2: Yes

5. Review Comments to the Author

Reviewer #1: The work is written in normal English. Possesses undoubted novelty and relevance. The work is based on an adequate bioinformatics analysis using standard algorithms in the R programming language and publicly available databases. However, the use of these algorithms and databases has allowed new data on compounds and mechanisms of their potential action in CRC to be obtained. This has important fundamental and applied significance.

Reviewer #2: The manuscript “Assessment of the targeted effect of Sijunzi decoction on the colorectal cancer microenvironment via the ESTIMATE algorithm “ by Jiaxin Du et al. describes the bioinformatic identification of potential targets of Sijunzi decoction (SJZD) that might play a role in colorectal cancer (CRC) via the regulation of the tumor microenvironment (TME).

The authors utilize publicly available gene expression and clinical data from two CRC patient cohorts to identify genes that are differentially regulated between tumors with high or low levels of immune and stromal cell infiltration, and then integrate these expression data with SJZD compound-target networks to identify TME-related genes, or their protein products, respectively, that are potentially targeted by SJZD active compounds, thus potentially explaining the beneficial effects of adjuvant treatment of CRC patients with SJZD.

However, there are numorous issues regarding methodology, data presentation and their interpretation that make the manuscript not acceptable for publication.

Major points :

1.

One major weakness is the rather limited number of patient samples analyzed in the present study. Could the authors validate their findings in an independent patient cohort (e.g. TCGA-COAD) ?

2.

In the methods section, the description regarding the correlation analysis of the ESTIMATE score are described inadequately. What exactly is the “correlation analysis model”, or a “correlation curve” and how was the survival analysis performed ?

3.

The relationship between immune cell and stromal infiltration and patient prognosis has been described previously. Hence, the analysis does not provide much additional insight.

Figure 1 A and B: These do not depict “correlation analyses”. It is unclear by which statistical method the p-values were obtained.

Lines 201 ff : It is not convincing to conclude any effects on survival, if the difference is not statistically significant.

4.

A description regarding the purpose of the GSEA and conclusions regarding its results are lacking. The results are not put into context with regard to the authors´research aim. Currently, the authors name a number of enriched gene sets without any interpretation of the results.

In the text, the authors refer to 613/48 and 730/41 gene sets in Figs 2 and 3, but only show a select few gene sets in the actual figure. Are these the statistically most significant gene sets ? Based on which criteria have they benn selected ?

Figures 2 and 3 do not have any figure legends. Normalized enrichment scores and statistical significance should be provided for each depicted gene set. It should be described in more detail which two groups were actually compared for the GSEA and how.

5.

Actually, the identification of DEGs is only described for normal vs tumor samples in the methods section. The identification of immune and stromal high / low DEGs should also be described, and lists with the identified DEGs should be provided.

Figures 4 A and B display identical heatmaps.

Figures 4 C,B : the labelling requires extensive editing

6.

What is the rationale to determine the overlap between potential SJZD targets and genes differentially expressed between normal and tumor tissues, since the authors identified immune and stromal high / low DEGs before ? This is confusing and should be explained more clearly.

Line 263 : Which DEGs were analyzed ? How were the 1607 genes in Fig 5B obtained ? This should be described explicitly and mentioned in the figure legend.

Line 264 : should be rephrased : “The active genes of SJZD”.

Furthermore, the statement “The active genes of SJZD also had differential expression in the tumor purity groups.” is not supported by any figure. Actually, this is also the more crucial analysis, since the authors aim to identify SJZD targets related to the TME, and not targets upregulated in tumor vs normal tissue.

Line 277: “The functional enrichment analysis of the DEGs” is actually not providing any information if the genes in the intersection (the 52 core genes) are related to “immune response”. Furthermore, a conclusion is lacking.

7.

Fig. 7: X-axis not properly labelled. A figure legend is missing.

Line 290 : “cytokine microarray”. Where is this shown ?

8.

Line 89 : The claim “we provided novel evidence that clarifies the influence of SJZD on the CRC TME” is an overstatement.

From this manuscript, it is actually not clear if SJZD has a defined (the authors only speak of "altered" expression in the discussion section) and measurable effect on the activity of any of the identified genes, or on the TME as such, in CRC.

Minor points :

1. Line 20 : please rephrase “Sijunzi decoction (SJZD) was used to treat patients with colorectal cancer (CRC) as an adjuvant method . The aim of the study was to investigate the effects of SJZD on the immune microenvironment of CRC”. The authors actually do not study the effects of SJZD on the tumor microenvironment, but rather aim to determine potential targets of SJZD related to the microenvironment using bioinformatic methods

2. Line 41 : delete “SJZD”

3. Lines 73f : please rephrase : “ could improve the survival and quality of life of patients with CRC by preventing tumorigenesis”

6. PLOS authors have the option to publish the peer review history of their article (what does this mean?). If published, this will include your full peer review and any attached files.

Reviewer #1: **Yes: **Denis S Kutilin

Reviewer #2: No

---

## [Author Response · Author response to Decision Letter 0]

13 Jan 2022

Dear editor,

Thank you! We are grateful to the reviewers for the valuable comments and suggestions. We also thank the editor for an opportunity to resubmit the manuscript (PONE-D-21-21615) after modifications. We have studied comments carefully and have made correction which we hope meet with approval. We listed the changes and marked in red in revised paper. Our point-by-point answers and explanations to the reviewers’ comments are as follows:

Question #1: One major weakness is the rather limited number of patient samples analyzed in the present study. Could the authors validate their findings in an independent patient cohort (e.g. TCGA-COAD) ?

Answer 1: Thank you for your suggestions! We performed further survival analysis validation by using 530 colon adenocarcinoma (COAD) samples from the TCGA database. The TCGA samples further confirmed the poor prognosis of HSPB1, SPP1, IGFBP3 and TGFB1. COL3A1 and OLR1 showed no significance in this progress, so the previous 6 core prognostic targets (HSPB1, COL3A1, SPP1, OLR1, IGFBP3 and TGFB1) were revised to 4 core prognostic targets (HSPB1, SPP1, IGFBP3 and TGFB1). Please see the Method (Line 188 - 193) and Result (Line 313 - 316) section.

Questions #2-3:

2.In the methods section, the description regarding the correlation analysis of the ESTIMATE score are described inadequately. What exactly is the “correlation analysis model”, or a “correlation curve” and how was the survival analysis performed?

3.The relationship between immune cell and stromal infiltration and patient prognosis has been described previously. Hence, the analysis does not provide much additional insight.

Figure 1 A and B: These do not depict “correlation analyses”. It is unclear by which statistical method the p-values were obtained.

Lines 201 ff : It is not convincing to conclude any effects on survival, if the difference is not statistically significant.

Answer 2-3: Sorry for our incorrect writing! We inappropriately described Figure 1 A and B as "correlation analysis models".We amended these sentences. Please see the part of methods (Line 119-122). For survival analysis, a more detailed introduction has also been added in the methods section (Line 124-133). In line 222 the sentence has been corrected as “while the patients showed no significant difference of OS between different immune groups”.

Question #4:

A description regarding the purpose of the GSEA and conclusions regarding its results are lacking. The results are not put into context with regard to the authors´research aim. Currently, the authors name a number of enriched gene sets without any interpretation of the results.

In the text, the authors refer to 613/48 and 730/41 gene sets in Figs 2 and 3, but only show a select few gene sets in the actual figure. Are these the statistically most significant gene sets ? Based on which criteria have they benn selected ?

Figures 2 and 3 do not have any figure legends. Normalized enrichment scores and statistical significance should be provided for each depicted gene set. It should be described in more detail which two groups were actually compared for the GSEA and how.

Answer 4: Thank you for your reminders. We added the method and purpose of GSEA in Line 135~143. Corresponding interpretation of the results can be seen in the results section "Results of Gene Set Enrichment Analysis". Enriched gene sets with FDR < 0.25, |normalized enrichment score (NES)| > 1, nominal P < 0.05 were regarded statistically significant [32]. The results of GSEA have been re-visualized and the top 5 gene sets (sort by NES) were shown in Figure 2. New figure legend has updated in figure 2, and reloaded online. More details of GSEA result were profiled in the S1 Appendix.

[32] Subramanian A, Tamayo P, Mootha VK, Mukherjee S, Ebert BL, Gillette MA, et al. Gene set enrichment analysis: a knowledge-based approach for interpreting genome-wide expression profiles. Proceedings of the National Academy of Sciences of the United States of America. 2005;102(43):15545-50. Epub 2005/10/04. doi: 10.1073/pnas.0506580102. PubMed PMID: 16199517; PubMed Central PMCID: PMCPMC1239896.

Question #5:

5.Actually, the identification of DEGs is only described for normal vs tumor samples in the methods section. The identification of immune and stromal high / low DEGs should also be described, and lists with the identified DEGs should be provided.

Figures 4 A and B display identical heatmaps.

Figures 4 C,B : the labelling requires extensive editing

Answer 5: Sorry for our incorrect writing! In fact, we didn't want to identify the DEGs between the normal and tumor samples. The incorrect description of "normal vs tumor" in the methods has been modified to "stromal high vs stromal low or immune high vs immune low" and more details have been added (Line169-176). Lists with the identified DEGs can be seen in S2 Appendix. The modified heatmaps and labels were shown in Figure 3 now.

Question #6:

6.What is the rationale to determine the overlap between potential SJZD targets and genes differentially expressed between normal and tumor tissues, since the authors identified immune and stromal high / low DEGs before ? This is confusing and should be explained more clearly.

Line 263 : Which DEGs were analyzed ? How were the 1607 genes in Fig 5B obtained ? This should be described explicitly and mentioned in the figure legend.

Line 264 : should be rephrased : “The active genes of SJZD”.

Furthermore, the statement “The active genes of SJZD also had differential expression in the tumor purity groups.” is not supported by any figure. Actually, this is also the more crucial analysis, since the authors aim to identify SJZD targets related to the TME, and not targets upregulated in tumor vs normal tissue.

Line 277: “The functional enrichment analysis of the DEGs” is actually not providing any information if the genes in the intersection (the 52 core genes) are related to “immune response”. Furthermore, a conclusion is lacking.

Answer 6: Thank you for your suggestions! The overlap between potential SJZD targets and stromal high / low DEGs was what we actually did. But we incorrectly described it as "overlap between potential SJZD targets and genes differentially expressed between normal and tumor tissues". This error has been corrected in answer 5. The origin of 1557 genes in Fig 5B was given in line 254-260. The incorrect sentences and figures have been rectified in the manuscript. The correct figures and the details of the DEGs have uploaded in the platform (Fig 3 & S2 Appendix).

Question #7:

Fig. 7: X-axis not properly labelled. A figure legend is missing.

Line 290 : “cytokine microarray”. Where is this shown ?

Answer 7: Sorry for our mistake. The statements of “cytokine microarray” was a clerical error and has been deleted. The correct figure has re-uploaded as Fig 6.

Question #8:

Line 89 : The claim “we provided novel evidence that clarifies the influence of SJZD on the CRC TME” is an overstatement. From this manuscript, it is actually not clear if SJZD has a defined (the authors only speak of "altered" expression in the discussion section) and measurable effect on the activity of any of the identified genes, or on the TME as such, in CRC.

Answer 8: Thanks for your kind suggestion. In Line 86, the statement has been adjusted that “The therapeutic targets of SJZD and the DEGs between different infiltration levels were intersected, and the core genes affected the TME of CRC were found. The result will provide the TME of CRC with a novel auxiliary therapeutic perspective.”

Question #9:

1. Line 20 : please rephrase “Sijunzi decoction (SJZD) was used to treat patients with colorectal cancer (CRC) as an adjuvant method. The aim of the study was to investigate the effects of SJZD on the immune microenvironment of CRC”. The authors actually do not study the effects of SJZD on the tumor microenvironment, but rather aim to determine potential targets of SJZD related to the microenvironment using bioinformatic methods.

2. Line 41 : delete “SJZD”

3. Lines 73f : please rephrase : “ could improve the survival and quality of life of patients with CRC by preventing tumorigenesis”

Answer 9: Thank you for your reminding. The objective of this study has been corrected in Line 21. In Line 71, the statement has been rephrased by “It has been widely reported that TCM could prolong the survival time and improve the quality of life of patients with CRC by preventing tumorigenesis, suppressing tumor growth, and reducing metastasis and recurrence”.

Kind regards

Xin-lin Chen

---

## [Editor Report · Decision Letter 1]

16 Feb 2022

Assessment of the targeted effect of Sijunzi decoction on the colorectal cancer microenvironment via the ESTIMATE algorithm

PONE-D-21-21615R1

Dear Dr. XIN-lin Chen,

We’re pleased to inform you that your manuscript has been judged scientifically suitable for publication and will be formally accepted for publication once it meets all outstanding technical requirements.

Kind regards,

Ilya Ulasov, Ph.D

Academic Editor

PLOS ONE

---

## [Editor Report · Acceptance letter]

10 Mar 2022

PONE-D-21-21615R1 

Assessment of the targeted effect of Sijunzi decoction on the colorectal cancer microenvironment via the ESTIMATE algorithm 

Dear Dr. Chen:

I'm pleased to inform you that your manuscript has been deemed suitable for publication in PLOS ONE. Congratulations! Your manuscript is now with our production department. 

Kind regards, 

on behalf of

Dr. Ilya Ulasov 

Academic Editor

PLOS ONE